# Risk Factors Associated With Mastitis in Smallholder Dairy Farms in Southeast Brazil

**DOI:** 10.3390/ani11072089

**Published:** 2021-07-14

**Authors:** Aline Callegari Silva, Richard Laven, Nilson Roberti Benites

**Affiliations:** 1Department of Preventive Veterinary Medicine and Animal Health, School of Veterinary Medicine and Animal Science, University of São Paulo (USP), Av. Prof. Dr. Orlando Marques de Paiva, 87, São Paulo CEP 05508-270, Brazil; benites@usp.br; 2School of Veterinary Science, Massey University, Palmerston North 4442, New Zealand; r.laven@massey.ac.nz

**Keywords:** clinical mastitis, subclinical mastitis, milking management, farm management

## Abstract

**Simple Summary:**

Bovine mastitis is a complex disease with many potential risk factors. However, few studies have reported the risk factors for mastitis in Brazil. This prospective, repeated cross-sectional study was carried out between May 2018 and June 2019 in smallholder dairy farms located in the southeast of Brazil. The potential risk factors for clinical and subclinical mastitis in smallholder dairy farms in Brazil were investigated by a combination of a questionnaire, clinical udder examinations and tests for subclinical mastitis. Risk factors for mastitis were evaluated at the cow level and at the herd level. The risk factors evaluated at the herd level were related to milking management, environment and management practices. We identified some risk factors; increased parity, later stage of lactation, not milking clinical and subclinical cases last, lack of routine cleaning of the milking parlor, using the dry-off treatment and optimized feed before calving. Further research on more farms across more areas of Brazil is required to develop targeted control programs for mastitis on smallholder dairy farms.

**Abstract:**

The aim of this study was to investigate the potential risk factors for clinical and subclinical mastitis in smallholder dairy farms in Brazil. A prospective, repeated cross-sectional study was carried out between May 2018 and June 2019 on 10 smallholder dairy farms. Potential risk factors for subclinical and clinical mastitis at the herd and cow level were recorded through interviewing the owner and by observation. A combination of clinical udder examination and the Tamis (screened mug with a dark base) test (Tadabras Indústria e Comércio de Produtos Agrovetereinário LTDA, Bragrança Paulista, SP, Brazil) were applied to observe clinical mastitis, and the California Mastitis Test (Tadabras Indústria e Comércio de Produtos Agrovetereinário LTDA, Bragrança Paulista, SP, Brazil) was used to determine subclinical mastitis. A total of 4567 quarters were tested, 107 (2.3%) had clinical mastitis, while 1519 (33.2%) had subclinical mastitis. At the cow level, clinical mastitis risk was highest in mid-lactation (50–150 days in milk) with OR 2.62 with 95% confidence interval (CI) of 1.03–6.67, while subclinical mastitis was highest in late lactation (> 150 days in milk) with OR 2.74 (95% CI 2.05–3.63) and lower in primiparous (OR 0.54, 95% CI 0.41–0.71) than multiparous cows. At the herd level, using dry-cow treatment (OR 4.23, 95% CI 1.42–12.62) was associated with an increased risk of clinical mastitis. Milking clinical (OR 0.37, 95% CI 0.24–0.56) and subclinical cases last (OR 0.21, 95% CI 0.09–0.47) and cleaning the milking parlor regularly (OR 0.27, 95% CI 0.15–0.46) had decreased odds for subclinical mastitis, while herds with optimized feed had greater odds (OR 9.11, 95% CI 2.59–31.9). Prevalence of clinical mastitis was at its lowest at the first visit in June/July and highest at the last visit in April/June (OR 3.81, 95% CI 1.93–7.52). Subclinical mastitis also presented increased odds in the last visit (OR 2.62, 95% CI 2.0–3.36). This study has identified some risk factors for mastitis on smallholder farms but further research on more farms across more areas of Brazil is required to develop a targeted mastitis control program for smallholder farms.

## 1. Introduction

Although Brazil is the world’s fifth-largest dairy producer, its dairy industry is principally made up of smallholder family-owned dairy farms (farms that are mainly operated by family labor, with up to four fiscal modules). These farms account for 81.2% of dairy farms in Brazil and are responsible for 64.2% of total Brazilian milk production [1]. This has impacts on animal health, especially mastitis, as the structure of the industry limits access to milk quality improvement programs, modern technologies and mastitis diagnosis [2]. The cow level prevalences of both clinical and subclinical mastitis are high in Brazil. In previous studies, Oliveira et al. [3] reported that ~ 30% of cattle had at least one case of clinical mastitis per annum, and that, including repeat cases, the average number of clinical mastitis cases per lactation was 1.02. In regard to subclinical mastitis, Busanello et al. [2] reported a prevalence of 46% with 18% of uninfected cows developing subclinical mastitis every month. However, none of these studies evaluated the risk factors of mastitis in smallholder dairy farms.

The high level of mastitis on Brazilian dairy farms results in significant economic loss, reduced animal welfare, milk quality and increased use of antibiotics [4,5].

Risk factors for mastitis can be divided into factors related to animals, and factors related to environment and management [6]. Previous studies which evaluated risk factors from mastitis found that the key risk factors at the cow level are age, lactation stage, milk production, breed and body condition score [3,7,8,9]. Key environmental and management risk factors include humidity, temperature, season, housing systems, herd size, milking management and management of the environment [5,10].

Few studies have presented herd characteristics and milk production systems in Brazil, however, Busanello et al. [2] using data from 517 herds, reported herd size was 82 (range = 11–1348), represented mainly by the southeast and presented an average production of 1597 kg/cow per year. Pathogens, such as *E. coli, Staphylococcus aureus* and *Streptococcus agalactiae* are still a challenge in most regions [2,3,5].

Few studies have reported the risk factors for mastitis on Brazilian dairy farms [3,9,11]. These studies have generally focused on larger farms. The aim of this study was to provide data on risk factors for mastitis on smallholder dairy farms in São Paulo state in Southeast Brazil.

## 2. Materials and Methods

All animal work was approved by The University of São Paulo ethics committee (Protocol no. 9901091216).

### 2.1. Description of the Study Area

The farms selected for the study were a convenience sample of dairy farms in the mesoregion of Piracicaba close to the University of São Paulo. The criteria for the selection of dairy farms were being a smallholder family farm, availability of good quality records and data, interest in participating in the research for at least one year and proximity to the University of São Paulo. A smallholder family farm is defined in Brazil as a farm less than four fiscal modules in size [12], that is mainly operated using family labor [13,14]. One fiscal module varies in each state of Brazil, in São Paulo state it is equivalent to 10 hectares (ha) of land [14] A total of fifteen farms (of which five were transitioning to organic production), were identified by technicians and extensionists linked to dairy activity in the region as being suitable for the study. From this initial group, ten farm owners (seven from conventional dairy farms and three transitioning to organic status) agreed to participate. All of the transitioning farms were in the first year of transition (a process that takes 18 months in Brazil [15]).

### 2.2. Data Collection

Farm visits—Data were collected between May 2018 and June 2019. All 10 farms were first visited in May/June 2018. Nine of the farms were then visited four times over the subsequent 12 months at 3–4 month intervals by the first author (a veterinarian). The remaining farm (which was a conventional farm) was visited for the second time in September 2018 but subsequently stopped producing milk so was not visited again.

Questionnaire—At the first visit, the farmers were interviewed about potential risk factors for mastitis at the herd and cow level using a prepared questionnaire (Appendix A). Prior to use in the study, the questionnaire had been tested on two farmers, who did not participate in the study. The questionnaire included fifty questions on general farm data (e.g., farm and herd size, and milk production), thirty-six questions on farm level management (e.g., feeding practices), and three questions on individual cow details (number, age and lactation stage). The questionnaire also contained six detailed questions on management measures related to mastitis, such as milking order of cows with confirmed mastitis (subclinical or clinical), teat disinfection pre- and post-milking, and use of gloves.

The milking procedure was then observed on each farm to confirm that the questionnaire answers were correct. During every visit, all milking cows were checked for clinical mastitis, using a combination of udder observation and palpation and the Tamis (screened mug with a dark base) test to observe changes in milk appearance [16], and for subclinical mastitis using the California Mastitis Test (CMT) [17]. Reactions were scored as zero or trace for negative, + 1, + 2, and + 3 for positive. For each case of mastitis (subclinical or clinical) the cow and affected quarter(s) were recorded.

### 2.3. Statistical Analysis

All statistical analyses were carried out using R version 3.1.1 (Rcore Team, 2014, Vienna, Austria). Two separate analyses were carried out: one with clinical mastitis as the dichotomous outcome variable at the quarter level and one with subclinical mastitis as the dichotomous outcome variable at the quarter level. For both analyses, data from quarters with the other type of mastitis were excluded from the model. Initially, for both clinical and subclinical mastitis, univariable multilevel logistic regression models (lme40 [18]) were created for all cow predictor variables at the cow level, and where there was variability between farms, at the farm level. A total of two predictor variables at the cow level were included, six milking practice predictor variables and seven environment and management predictor variables at the herd level. Farm and cow were included as random effects in these models. Predictors with *p*-value < 0.25 were then put forward for testing in multivariable models [19]. Potential predictors were assessed for collinearity and not included in the further models when the correlation was > 0.6, where there was collinearity the variable with the lower *p*-value was selected for inclusion in the modeling process. The explanatory variables related to farm structure (e.g farm size, feeding practice, number of cows) were not included in the models.

Three models were created for each outcome variable—cow variable model (parity, lactation stage), herd milking practices variables model (milking clinical cases last, milking subclinical cases last, use separate clothes, gloves, disinfecting cluster between milk, cleaning milk parlor regularly) and environment and management model (production system, visit date, protocol treatment by a veterinarian, dry-cow treatment, J5 vaccine, dry cow minerals, homeopathic salt, optimize feed before calving) (Table 1).

In the cow variable model, the lactation stage was categorized as < 50 days in milk (DIM), 50–150 DIM and > 150 DIM. Categorization of DIM was limited to these three categories because some farmers had no exact information on the DIM of their cows.

For the herd variable model, washing dirty teats, replacing teat cup liners every 6 months, washing milking liners at the end of milking and keeping cows standing were not included in the model as they were used on all farms. Pre- and post-dipping were not included because only two farms did not use these methods and those two farms allowed calves to suckle from the udder before and after milking.

A backward stepwise approach was used; the variable with the highest *p*-value was removed until all ramming variables had a *p*-value < 0.05. To check for confounding, the variables removed during the initial process were added back in one by one. A variable was considered as a confounder if its removal resulted in changes of the remaining predictors of ≥ 20%. Confounders were kept in the model.

## 3. Results

### 3.1. Farm Description

All farms kept their cattle in a confined or semi-confined housing system, one of the farms had no access to grazing, whereas the other nine had limited access to grazing. Main feeds were silage (mean 26.5 kg/cow/day; range 15–35 kg/cow/day) and concentrate (mean 3.8 kg/cow/day; range 0.2 to 6 kg/cow/day).

A range of breeds were present on all farms, with Holstein (52%) and Gyr × Holstein crossbreeds (40%) being the commonest breeds. The remaining cattle were purebred Jersey (8%).

Mean farm size was 12.6 ha (range 5–24 ha), mean lactating herd size was 33 cows (range 12–46 lactating cows) and average total herd milk production was 13.6 kg milk/day/cow (range 8–20.5 kg milk/day/cow). These figures were similar to those reported by Balcão et al. [20] in their survey of smallholder farms in Brazil. Bulk milk somatic cell count (SCC) was not included in this study.

### 3.2. Clinical and Subclinical Mastitis Prevalence

In total, 1165 cows and 4567 udder quarters were examined according to mastitis status (clinical, subclinical or absent) during the study period. Overall, 83 (7.1%) cows had at least one quarter with clinical mastitis and 704 (60.4%) had at least one quarter with subclinical mastitis. Of the 4567 quarter observations, 107 (2.3%) had clinical mastitis, while 1519 (33.2%) had subclinical mastitis. Overall, a total of 67 cows had both clinical mastitis (CM) and subclinical mastitis (SC).

The mean herd level prevalence of clinical mastitis was 2.26 (range 0–9.27 observations) and for subclinical mastitis was 32.50 (range 8.33–59.68). The mean of 2 observations (range 1–4 observations) was made in the same cow.

### 3.3. Risk Factor Analysis

The parity and lactation stage remained in both the clinical and subclinical mastitis models. Compared to multiparous cows, first parity cows had half the odds of having clinical and subclinical mastitis (Table 2). The odds ratio for both clinical and subclinical mastitis was higher for cows in later stages of lactation compared to cows in earlier lactation (< 50 days in milk (DIM)) (Table 2).

At the herd level, the model for milking management practices and clinical mastitis failed to converge, so no multivariate analysis was possible. Table 3 summarizes the explanatory variables included in the final multivariable model of subclinical mastitis and milking management practices. Of these factors, all three (milking clinical or subclinical cases last, and regularly cleaning the milking parlor) were associated with reduced odds of subclinical mastitis (Table 3).

Table 4 summarizes the models investigating the association of herd level environment and management factors with clinical and subclinical mastitis. For the clinical mastitis model, the variables remaining were dry-cow treatment, vaccination with a J5 vaccine, and visit date. Using dry-cow treatment or vaccination were both associated with increased odds of clinical mastitis, although the confidence intervals were wide especially for vaccination. The subclinical mastitis model included vaccination, optimizing feeding and visit date. In contrast to clinical mastitis, vaccination was associated with a reduced risk of subclinical mastitis, although the confidence intervals were wide. Prevalence of both clinical mastitis and subclinical mastitis were at their lowest at the first visit in June/July and highest at the last visit in April/June. The production system (conventional or transitioning to organic) did not remain in the model.

## 4. Discussion

This is, as far as the authors are aware, the first study of the risk factors associated with mastitis (clinical or subclinical) that has focused on smallholder dairy farms in Brazil. In addition, many previous studies of risk factors for mastitis in tropical smallholder dairy farms, e.g., [17,21] have not been longitudinal studies with multiple measurements on the same farms. Thus, although relatively small and a convenience sample (which limits the generalizability of the results), this dataset does provide useful information on the risk factors for mastitis in an understudied part of the Brazilian dairy industry. Further research is clearly needed but such research will have to address the issues with lack of good quality data on most Brazilian smallholder dairy farms and the difficulty of maintaining the interest of farm owners, which greatly limited the number of farms included in this study.

The proportion of quarters identified as having clinical mastitis over the period of the study was 2.3%, with, at the cow level, 7.1% of observations identifying clinical mastitis in at least one quarter. This is not directly comparable to the figures reported by Oliveira et al. [3] as the data from the current study are average prevalence at four examination points rather than lactational incidence but the figures from this study are compatible with the very high mean lactational incidence (~100%) reported by Oliveira et al. [3]. The 2.3% figure for quarter level clinical mastitis is higher than the 1.3% reported in smallholder farms in southern Ethiopia by Abebe et al. [22], and the 7.1% reported at the cow observation level is higher than the 4.8% figure reported (at the cow level) in smallholder farms in Zimbabwe [23].

The proportion of quarters identified as having subclinical mastitis over the period of the study was 33%, with 60.4% of cow observations identifying at least one affected quarter. This is not directly comparable to the figure reported by Busanello et al. [2] of 46% at the cow level, as that was a single point prevalence. However, it is compatible with that figure and could indicate even higher rates of subclinical mastitis in these farms than reported by Busanello et al. [2] in larger Brazilian dairy farms. The 33% of quarters with subclinical mastitis reported in this study is very similar to the prevalence of subclinical mastitis at the quarter level reported in smallholder farms in other comparable countries, for example, the 36% reported by Abebe et al. [22], 33% reported by Mekonnen et al. [24] and 37% reported by Ndahetuye [21], all of whom reported cow level prevalences of 62%.

Larger scale studies across more farms in more districts are required to better characterize mastitis risk in smallholding dairy farms in Brazil, but our results suggest that clinical and subclinical mastitis may be as much a problem on Brazilian smallholdings as they are on larger dairy farms and maybe even worse.

At the cow level, parity and lactation stage were the only risk factors included in the final model for both clinical and subclinical mastitis. An increased risk of mastitis for multiparous compared to primiparous cows is a consistent effect found in many studies of risk factors for mastitis, e.g., [3,9,25] including those that have focused on smallholders, e.g., [10,22,24]. For both clinical and subclinical mastitis, later lactation stages were associated with higher odds of occurrence. This is consistent with many previous studies looking at mastitis in smallholder dairy farms, e.g., [10,17,21,24] though not all, e.g., [22] and also studies in larger farms, e.g., [9] The effect of lactation stage on subclinical mastitis is likely to be related to the accumulation of chronic infections which have not been identified during lactation. In contrast, the reason why clinical mastitis was more common in later lactation is less clear, especially as in larger dairy herds, clinical mastitis appears to be more common in early lactation [3,26].

We were not able to identify any associations between herd level milking management practices and clinical mastitis because the model did not converge, but for subclinical mastitis the practices remaining in the final model were milking clinical cases last, milking subclinical cases last and regularly cleaning the milking parlor. All of these practices have been associated with reduced mastitis in large commercial dairy herds [9,27,28], and on smallholdings, e.g., [22,29]. The number of milking management practices that this study identified as being associated with subclinical mastitis in this study is relatively low, but this may be because the study lacked power as only 10 farms were included in the study and, for many potentially important practices, there was very little variation between farms. Thus, the absence of management practice in the final model should not be taken to indicate that practice is not a risk factor for subclinical mastitis.

The herd level environment and management practices, which remained in the final model for clinical mastitis, were dry-cow therapy, use of a J5 vaccine and time of year. The first two practices were both associated with increased risk of clinical mastitis despite both being intended to reduce clinical mastitis [30,31]. Details were not collected on either the vaccination or the dry-cow regime, but it is likely that the positive association was because farms that used vaccination or dry-cow therapy were responding to a clinical mastitis problem so were likely to have a higher prevalence of mastitis than farms which did not use vaccination or dry-cow therapy. This paradoxical effect may have been exacerbated by incorrect use of the treatments (e.g., poor timing of vaccination or inadequate hygiene during dry-cow therapy) which would have possibly resulted in them having no or limited impact on mastitis. Additionally, the poor management of the herd could contribute to the re-infection of cured cows [32]. Subsequent studies should look more closely at the use of these therapies by smallholders and identify why and how they are being used.

Clinical mastitis was at its lowest prevalence at the first visit in June/July and highest in the last visit in April/June. In southeast Brazil, there are two seasons: the cool dry season (April–September) and the warm rainy season (October–March) [33]. June to August are the coldest and driest months with mean temperature and rainfall of 15.4 °C, and 35 mm respectively. January is the hottest and driest month with a mean temperature and rainfall of 23.6 °C and 218 mm (data from the INMET; Instituto Nacional de Meteorologia, Brazil; www.inmet.gov.br, accessed on 12 October 2020). The lower risk in the first visit (June/July) and the high risk of mastitis in the third visit likely reflect environmental mastitis [3]. However, increased CM odds in the fourth visit could have been influenced by pathogen persistence in the herd, associated with heat stress until April–May [5]. Subclinical mastitis was also apparently affected in a similar way. Further investigation of the role of the season in mastitis on smallholder farms in Brazil may assist the development of targeted control programs for use at high-risk periods of time.

In addition to the season, vaccination with a J5 vaccine and the optimization of feed were also in the final model relating herd-level environment and management practices to the prevalence of subclinical mastitis. It is unclear why J5 vaccination was related to reduced prevalence of subclinical mastitis as that vaccine targets *E. coli* which is not an important cause of subclinical mastitis [30]. Optimization of feed was associated in this dataset with a very large increase in the odds of a quarter that had subclinical mastitis (OR 9.1). This may reflect that farmers who optimize feed have generally higher production cows compared to other farms in the study, which may increase mastitis risk [34]. Further information is required on the association between feeding, production level, and the risk of mastitis (both clinical and subclinical) on smallholder dairy farms in Brazil.

No effect of farm system on mastitis risk was observed in this study. This suggests that smallholder farms that transition to organic farming do not necessarily have to have increased mastitis risk, but further research is required to confirm that this will be the case on most farms.

## 5. Conclusions

This study in 10 smallholder dairy farms in Brazil found a high prevalence of both clinical and subclinical mastitis. This study has identified some risk factors for mastitis on such farms but further research on more farms across more areas of Brazil is required to develop targeted control programs for mastitis on smallholder dairy farms. This research should be combined with identifying the knowledge gaps of smallholders in regard to mastitis and the barriers to implementing mastitis control on such farms.

## Figures and Tables

**Table 1 animals-11-02089-t001:** Selected predictors included in the univariable models for cow and herd level analyses as risk factors for clinical and subclinical mastitis in lactating cows assessed in a longitudinal study on smallholder dairy farms (*n* = 10) located in the southeast of Brazil.

Variable	Unit/Category		Clinical Mastitis	Subclinical Mastitis
			Yes/No	Yes/No
Parity	1		15 (1.3%)/1133 (98.7%)	285 (25.2%)/848 (76.8%)
	≥ 2		91 (2.7%)/3284 (97.3%)	1222 (37.2%)/2062 (62.8%)
Lactation stage	< 50 DIM *		10 (1.6%)/621 (98.4%)	141 (22.7%)/480 (77.3%)
	50–150 DIM		33 (2.7%)/1203 (97.3%)	360 (29.9%)/ 843 (70.1%)
	> 150 DIM		49 (2.2%)/2186 (97.8%)	876 (40.1%)/1310 (59.9%)
Milking practices		Herd (no.)		
Milking clinical cases last	Yes No	6 4	23 (1.4%)/1653 (98.6%) 84 (2.9%)/2807 (97.1%)	339 (20.5%/1314 (79.5%) 1180 (42.0%)/1627 (58.0%)
Milking subclinical cases last **	Yes No	9 1	0 (0%)/264 (100%) 107 (2.5%)/4196 (97.5%)	22 (8.3%)/242 (91.7%) 1497 (35.7%)/2699 (64.3%)
Use separate clothes	Yes No	3 7	93 (2.6%)/3489 (97.4%) 14 (1.4%)/971 (98.6%)	1242 (35.6%)/2247(64.4%) 277 (28.5%)/694 (71.5%)
Gloves for milking	Yes No	8 2	27 (2.1%)/1254 (97.9%) 80 (2.4%)/3206 (97.6%)	406 (32.4%)/848 (67.6%) 1113 (34.7%)/2093 (65.3%)
Disinfecting cluster between milk	Yes No	7 3	49 (2.8%)/1716 (87.2%) 58 (2.1%)/2744 (97.9%)	631 (36.8%)/1085 (63.2%) 888 (32.4%)/1856 (67.6%)
Cleaning milk parlor regularly ***	Yes No	1 9	69 (64.5%)/38 (36.5%) 4088 (91.7%)/372 (8.3%)	1297 (31.7%)/2791 (68.3%) 222 (59.7%)/150 (40.3%)
Environment and management				
Production system	Conventional Organic’s transition	7 3	78 (2.7%)/2814 (97.3%) 29 (1.7%)/1646 (98.3%)	961 (34.2%)/1853 (65.8%) 558 (33.9%)/1088 (66.1%)
Visit date	First Second Third Fourth	10 10 9 9	16 (1.3%)/1173 (98.7%) 17 (1.4%)/1231 (98.6%) 29 (2.8%)/1021 (97.2%) 45 (4.2%)/1035 (95.8%)	321 (27.4%)/852 (72.6%) 447 (36.3%)/784 (63.7%)) 345 (33.8%)/676 (66.2%) 406 (39.2%)/629 (60.8%)
Dry-cow treatment	Yes No	5 5	78 (3.2%)/2362 (96.8%) 29 (1.4%)/2098 (98.6%)	897 (38.0%)/1465 (63.0%) 622 (29.6%)/1476 (70.4%)
J5 vaccine	Yes No	8 2	37 (3.1%)/1142 (96.9%) 70 (2.1%)/3318 (97.9%)	487 (42.6%)/655 (57.4%) 1032 (31.1%)/2286 (68.9%)
Dry cow minerals	Yes No	9 1	23 (3.6%)/623 (96.4%) 84 (2.1%)/3837 (97.9%)	274 (44.0%)/349 (56.0%) 1245 (32.5%)/2592 (67.5%)
Homeopathic salt	Yes No	3 7	69 (1.9%)/3636 (98.1%) 38 (4.4%)/824 (95.6%)	1233 (33.9%)/2403 (66.1%) 286 (34.7%)/538 (65.3%)
Optimize feed before calving (15days) ****	Yes No	7 3	75 (4.7%)/1514 (95.3%) 32 (1.0%)/2946 (99.0%)	709 (46.8%)/805 (53.2%) 810 (27.5%)/2136 (72.5%)

* DIM = Days in milk. ** Milking subclinical cases last: identified by California Mastitis Test (CMT). *** Clean milk parlor regularly: refers to milk parlor being cleaned after every milking. **** Optimize feed before calving: refers to prepartum cows fed with supplementary diet 2 weeks before calving.

**Table 2 animals-11-02089-t002:** Final multivariable logistic regression model of associations between cow level risk factors and clinical and subclinical mastitis at a quarter level in 10 smallholder dairy herds in Piracicaba Mesoregion, São Paulo, Brazil.

Variable	Category	Odds Ratio (95% CI)	Odds Ratio (95% CI)
		Clinical Mastitis	Subclinical Mastitis
Parity			
	multiparous	Reference	Reference
	primiparous	0.54 (0.26–1.13)	0.54 (0.41–0.71) ***
Lactation stage			
	<50 DIM	Reference ***	Reference
	50–150 DIM	2.62 (1.03–6.67) *	1.62 (1.19–2.18) **
	>150 DIM	1.83 (0.75–4.48)	2.74 (2.05–3.63) ***

* DIM = Days in milk. * *p* < 0.01. ** *p* < 0.001. *** *p* < 0.0001.

**Table 3 animals-11-02089-t003:** Final multivariable logistic regression model of associations between herd level milking management practices and subclinical mastitis (positive California Mastitis Test) at a quarter level in 10 smallholder dairy herds in Piracicaba Mesoregion, São Paulo, Brazil.

Variables	Category	Odds Ratio (95% CI)
Milking clinical cases last	No Yes	reference 0.37 (0.24–0.56) ***
Milking subclinical cases last	No Yes	reference 0.21 (0.09–0.47) ***
Clean milk parlor regularly	No Yes	Reference 0.27 (0.15–0.46) ***

*** *p* < 0.0001.

**Table 4 animals-11-02089-t004:** Final multivariable logistic regression model of associations between herd level environmental and management practices and clinical and subclinical mastitis at a quarter level in 10 smallholder dairy herds in Piracicaba Mesoregion, São Paulo, Brazil.

Variable	Category	Odds Ratio (95%CI)	
		Clinical Mastitis	Subclinical Mastitis
Dry-cow treatment	No Yes	Reference 4.23 (1.42–12.62) ***	-
Vaccine	No Yes	Reference 2.44 (0.74–7.97)	Reference 0.32 (0.07–1.33)
Optimize feed before calving	No Yes	-	Reference 9.11 (2.59–31.9) ***
Visit date	First (June–July)	Reference	Reference
	Second (September)	1.28 (0.60–2.73)	2.07 (1.65–2.60) ***
	Third (December–January)	2.56 (1.25–5.24) **	1.82 (1.42–2.33) ***
	Fourth (April–June)	3.81(1.93–7.52) ***	2.62 (2.04–3.36) ***

** *p*<0.001. *** *p*<0.0001. - Variable not remaining in the final multivariable model.

## Data Availability

The data presented in this study are available within the article.

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
