# Peer review of "Risk Factors Associated With Mastitis in Smallholder Dairy Farms in Southeast Brazil"

_animals, 2021, doi:10.3390/ani11072089_

Round 1

Reviewer 1 Report

The study “Risk Factors Associated With Mastitis in Smallholder Dairy Farms in Southeast Brazil” enters the topics of the journal, it is of great interest to readers, and I believe that it could contribute to development of public policies and improvement of dairy production

In this research risk factors for subclinical and clinical mastitis in smallholders dairy farms located in the southeast of Brazil were identified. The factors tested were: parity, lactation stage, milking clinical and subclinical cases last, clean milk parlor, dry off treatment, optimize feed.

This study is great interest because it was focused to research risk factors for subclinical and clinical mastitis in smallholders dairy farms located in the southeast of Brazil. More of them, have generally focused on larger farms. Smallholders need academic support to improve their viability in several fields. Nevertheless, given the small sample size, it seems to correspond more to an exploratory study (short communication) than a full article.

I have some questions and suggestions that I would like to point out to you. Please consider them as a way to improve the manuscript.

Introduction

*According to Rangel el at (2021) in Animals journal you can found wide information about smallholders in tropical zones.

Material and method

The criteria for selection of  dairy farms were being a smallholder family farm (farm size up to 40 ha), availability of good quality records and data, interest in participating in the research for at least one year  and proximity to the University of São Paulo.

These criteria make the sample oriented to medium-sized farms, rather than to smallholders (medium size, good records, etc...)

*Data collection

The sample (insufficient) is also very heterogeneous both in the collection of the information, as well as the conventional and organic production. It does not respond to a clear criterion. This part could be removed (line 79-81).

Date  from 10 farms were collected ....

* Regards to selected predictors included in the univariable models:

Why these? Why have structural indicators been excluded?

Results

There are a strong heterogeneity of the facilities, feeding (Main feeds were silage 134 (mean 26.5 kg / cow / day; range 15-35 kg / cow / day) and concentrate (mean 3.8 kg / cow / day; 135 range 0.2 to 6 kg / cow / day) and other factors. Allow me to recommend an increase in the sample size so that the results will be robust and consistent.

The results were significant. The number of factors considered is reduced. The excluded factors were not important? or do we not have sufficient information for the results to be consistent?

It is very clarifying:

We were not able to identify the association between herd level milking management  practices and clinical mastitis because the model did not converge

The aim of this study was to provide data on risk factors for mastitis on smallholder dairy farms in São Paulo state in SE Brazil.

Has the objective been achieved?

The results obtained were representative of smallholders dairy farms in Sao Paulo state in SE Brazil?

I invite you to argue your application or its non-application.

Reviewer 2 Report

General Comments to Authors:

This is a generally well written paper that describes a small scale study. The results will be of interest to some readers, but the small number of enrolled farms and the lack of detail regarding some of the study design (especially the lack of detail on the questionnaire itself) limits the relevance to many readers.

The paper would be much stronger of the authors provided more detail on the methodology of the study, and perhaps placed more emphasis on interpreting their findings rather than reporting results.

I am concerned that the study design has the potential for considerable bias because some potentially important risk factors may have been ignored or overlooked in drawing data from so few herds. It would be valuable for the authors to acknowledge these limitations and discuss them.

Specific Comments to Authors:

L11:           “The bovine mastitis is a complex…” should be “Bovine mastitis is a complex…” This is an example of simple errors in grammar that occur throughout the manuscript. Whilst they do not alter the scientific merit of the study, they interfere with the readability of the manuscript and need to be addressed.

L28:           “At the herd level, using drying off.. …associated with CM risk.” could be “associated with increased risk of clinical mastitis.” Specifying the direction of the relationship makes the message a little easier to grasp for the reader.

L51:           In this paragraph, it is not immediately obvious that the authors are referring to risk factors for mastitis beyond the local Brazilian system. A little more explanation would make things clearer.

L56:           Delete “Relatively”

L57:           The use of “e.g.” prior to lists of references is inappropriate.

L69:           “Brazil a farm” ??

L69:           The meaning of “4 fiscal modules” needs to be explained.

L72:           “…as farm suitable…” ??

L92:           Who checked the cows for clinical mastitis? Was it the researcher, the milking staff or somebody else? What efforts were made to ensure consistency of diagnosis?

L118:         The variation in milking management between enrolled herds is a significant limitation to the analysis of this data. In such a small study, ignoring factors which may have substantial effects on the occurrence of mastitis (eg teat dipping, allowance of calf suckling) is likely to introduce considerable potential for bias. This needs to be addressed by the authors. It may not be possible to accommodate in the analysis, but it should be acknowledged and the possible effects on the study’s conclusions discussed.

Table 1:   The 4th column does not provide a “Distribution”, it provides a proportion. The Table could be simplified to provide the same data but in a less extensive table.

L133:         For readers who are not familiar with Brazilian farming systems, “semi-confined housing” should be more clearly explained.

L135:         Where the concentrates which were being fed in the form of grain, or commercial pellets, or something else? Is it possible that the amount of concentrate being fed, and the type, might be a factor for the occurrence of mastitis (due to effects on transition cow nutrition and early lactation production levels)?

L203:         Is it valid to compare these findings with the findings of Oliveira, and those reported for Ethiopian and Zimbabwean studies? Are the production systems for smallholder farms similar in Brazil, Ethiopia and Zimbabwe? Are the farming systems studied by Oliveira comparable to those of the current study?

L242:         What does “having a clean milk parlour” mean? How were these factors defined, and how was the classification of farms into such groups kept consistent? Where possible, the criteria should be explained.

L253:         The authors recognise that a limitation of establishing associations is that they do not provide evidence of causality. Was the use of dry cow or J5 vaccination a risk factor that can lead to mastitis, or a consequence of the farmer’s attempt to respond to a mastitis problem that existed already? The authors make a very superficial attempt to address this aspect of their study, but it is an important aspect and worthy of considerably more emphasis and explanation than is given.

Reviewer 3 Report

Dear authors,

Thanks for an interesting paper about risk factors for mastitis in smallholder herds in Brazil. Although I think the manuscript have merits, I have some questions and suggestions.

A general question – why did you choose to use your definition of smallholder herd? Why not use herd size? To my knowledge not many studies about mastitis use farm size in ha as selection, it more common to use herd size. It would also have been easier for you to compare your results with others. If you consider herd size the participating herds are not so small compared to e.g. dairy herds in Rwanda. Moreover, all herds in this study seemed to use machine milking which might be a difference from other smallholder herds.

Please add a final model including all risk factors offered to the three models.

Please include a supplement/appendix with the distribution of CM and SCM for all the answers to the questionnaire including p-values. This information is useful for others that want to do a similar study or just compare their results with yours. Moreover, the answers will give the reader the opportunity to get a picture of how smallholder dairy herds in Brazil are managed and if the lack of finding important risk factors is due to that all the participating herd hade very similar management. If all herds do things that previous studies have shown to be increasing the risk of mastitis that knowledge could help you make a control program.

Just a question but why did you ask so many questions about costs? Is that for another paper?

Simple summary

L12: ”these diseases” – which diseases do you mean? Or should it say this disease (bovine mastitis)?

L15: change “the questionnaire” to “a questionnaire”, change “udder clinical examination” to “clinical udder examinations”

L15-16: change “subclinical test” to “subclinical tests”

L16-17: consider the wording

Abstract

L22-23: The abbreviations CM and SCM is not used later in the manuscript, please consider this and be consistent throughout the text.

L24: change “udder clinical examination” to “clinical udder examinations”

L25-26: please use decimal when presenting numbers>999, i.e. 4,567 and 1,519

L28: please change “drying off treatment” to “dry-cow treatment”

L29: a space is missing after the dot

L33-35: As you found several risk factors similar to what other studies have found I think it would be better to develop a control program and test the effect of applying it, you always have to adopt the actions to the individual farms prerequisites not all risk factors are present or as much risk in all farms. Consider this in your statement in these lines.

Keywords

L36: A suggestion is to use words not included in the heading, that will increase the “hits”, so maybe change risk factor and mastitis to clinical mastitis and subclinical mastitis.

Introduction

L40: What is the definition of smallholder in this sentence?

L44-48: What type of herds were included in Oliviera’s and Busanello’s studies?

L52-55: if you already know all these key risk factors why did you need to perform your study? Please rephrase the sentences e.g. “In previous studies xxx have been identified as risk factors, however…”

L53: Is season really a cow level factor?

Please add some information about dairy farms in Brazil, average herd size, average milk production, BMSCC and or cow SCC, common bacterial findings at CM and SCM (i.e. is contagious mastitis or environmental mastitis most common).

Material and Methods

Data collection

L66 vs. L68-70: Please consider removing (farm size ut to 40 ha) as it gets confusing when you read the sentence in L68-70 – is 40ha the same as 4 fiscal modules? Or combine the sentences more.

L68-70: How many smallholder farms are there in Brazil and or in the mesoregion of Piracicaba?

Data collection

L78: a space to much after the dot

L82: How long time took the interview to perform?

L85-90: Could you please include number of questions for each subject of question (number of questions concerning general farm data, farm level management etc.)?

L91-92: please add that this also was only done at the first visit, and perhaps add that if not the answers were correct, they were changed accordingly? In how many cases did this differ?

L95: What CMT score was used to define a quarter as having SCM?

Statistical analysis

Why did you not perform a final multivariable model? Especially it would be of interest to see how the results changed if you included the cow factors as they usually explain most of the variation seen. Although it is difficult to do anything about the cow factors in practice, we might to give more advice in taking extra care of the cows in different periods of lactation/ages.

Why did you categorize DIM as you did? In most studies the risk/odds of CM is highest around calving and up to about two weeks after calving so it would have been logical to have more categories or use DIM as a continuous variable (if it had a linear relationship with the outcomes).

L110-115: Could you please include number of variables for each subject of question (number of variables included in the cow level model to start with etc.)? Consider using cow/herd/environmental and management variable model instead of cow/herd/environmental and management level model as the observational level is quarter level.

Table 1: Heading: please consider changing “…cow- and herd level analysis…” to “…cow- and herd level variables associated with presence of clinical or subclinical mastitis at quarter level…” Please include distribution of CM and SCM for each category.

L126-127: Please considering changing “parameters” to “variables” for consistency and to change to “Using a backward stepwise approach, variables with a p-values <0.05 were kept in the multivariable models”

Results

Farm description

Did you not have any information about BMSCC? Did the information on breed and milk production only consist of herd level data?

Clinical and subclinical mastitis prevalence

L147-150: Please consider rewriting this section for clarity e.g. In total, 1165 cows and 4567 udder quarter were examined according to mastitis status (clinical, subclinical or absent) during the study period. Overall, 83 (7.1%) of the cows had at least one quarter with clinical mastitis and 704 (60.4%) had at least one quarter with subclinical mastitis.

How many of the cows had both CM and SCM? Overall and at same visit?

How many observations per cow were made (median and min-max)?

What was the herd level prevalence?

Risk factor analysis

Heading: please change F to f.

L154-155: Consider combining the results for CM and SCM here, e.g. Both parity and lactation stage remained in both the model of associations with having clinical, as well as subclinical mastitis.

L155-157: Please consider rewriting this section for clarity, e.g. First parity cows had half the odds of having clinical, as well as subclinical, mastitis compared to multiparous cows (Table 2). Or if you want to make statements about the OR for multiparous cows - change reference in the table so the primiparous cows are the reference. Moreover, it is not necessary to include OR and CI in the text if you use multiparous cows as reference and use the “First parity had half the odds…” sentence, making it more similar to next sentence (where OR and CI is not given). Also, as it is written now referring to table 2 after giving the OR and CI is not correct as those numbers are not included in table 2…

L157-158: Please consider rewriting this section for clarity, e.g. The odds ratio for both clinical and subclinical mastitis was higher for cows in later stages of lactation compared to cows in earlier lactation (<50 days in milk (DIM)).

Table 2. Heading – please change …model of the association… to …model of associations…, also please change …at the quarter level to …at quarter level. Include number of observations (quarters) used in the final model. Please add p-values for each variable and category (you can move the categories to the column “variable” to make more room).

L164-168: consider removing if you combine the results according to suggestion (L154-155).

What cow factors were registered other than parity, age, and lactation stage?

L169-170: Did you investigate which variable that caused the non-convergence? Perhaps you could have performed a multivariable analysis just by excluding some of the variables? To my experience it is usually one or two variables with few observations in some category that causes the non-convergence. As herd were included as random factor perhaps “milking subclinical cases last” and/or “Clean milking parlor” caused the non-convergence?

Table 3. Heading – please change …model of the association… to …model of associations…, also please change …at the quarter level to …at quarter level. Include number of observations (quarters) used in the final model. Please add p-values for each variable and category (you can move the categories to the column “variable” to make more room).

L179: Please change “…to clinical…” to “…associated with…”

L179-180: please change “…the variables included…” to “…the variables remaining…” as more variables were included to start with but those remained after the backward elimination.

L186: You do not mention visit date results in the text, why? What about breed, milk production, herd size, farm size – were they investigated in the univariable/multivariable models? If not – why?

Table 4. Heading – please change …model of the association… to …model of associations…, also please change …at the quarter level to …at quarter level. Include number of observations (quarters) used in the final model. Please add p-values for each variable and category (you can move the categories to the column “variable” to make more room). Moreover, there is an row with just “-“ for both CM and SCM – please remove that row.

Discussion

L192-202: Why would the risk factors for CM and SCM be so different between smallholder farms and larger farms? You asked about the same questions as you would in a larger herd, didn’t you? What was special for smallholder dairy herds? How did the longitudinal part of the study contribute with more information beside giving more observations?

I also thinks more research is needed but then more importantly on how to get the farmers to do what we think they should do in order to reduce the incidence/prevalence of mastitis (how to best give advices in order to get changes) and to do intervention studies so we can say what risk factor would contributes most to a reduced risk of mastitis. Also, if the bacterial panorama is unknows it would be of importance to investigate that in order to be able to design a control program – is it more common to have contagious mastitis or environmental and dose it vary between herds?

L208: I do not understand how 2.3% or 7.1% is comparable to 100% - please clarify.

L209-212: Is smallholder farms in Ethiopia and Zimbabwe similar to smallholder farms in Brazil? Climate, herd sizes, breeds, milk yield? Could that explain the difference in prevalence (although the difference is so small, I would say it is very similar…) – please motivate why you compared it with those studies.

L218-222: Please change the beginning of the sentence as you now interpret it as an example of larger Brazilian dairy farms, perhaps start with “In smallholder farms in other comparable countries the prevalences of subclinical mastitis at quarter and cow level were similar to the present study; 36% and 33% in Ethiopia (Abebe et al. and Mekonnen et al.) and 37% in Rwanda (Ndahetuye et al.), which all reported at cow-level prevalence between 59.2-62%.  

L227: Please considering changing “included” to “remaining” if more variables were included to start with in the multivariable model.

L236-237: This statement is not for only larger Brazilian dairy herd, but a finding in many other studies. Please include one or two more references on this. Please discuss the choice of categories for DIM – could that affect your results?

L239: please change “…the association…” to “…any associations…”

L246: Did the study lack power when you had more than 4,000 observations? I believe it is more the lack of variation. Also, if it is more common with environmental mastitis the milking management practices might be of less importance compared to cleanliness (to reduce the bacterial load) and proper feeding (to enhance the immune system). Did you assess the cleanliness of the housing and animals when you made your observations? Did you register the body condition scores?

L257-260: Or, the solution is not giving treatments or vaccine but removing the risk factors by preventive actions! If these farmers do nothing to minimize the bacterial load and improve the cow’s immune status but dry-treat and/or use vaccine it is very likely they won’t succeed.

L261: Have no other studies found this? Have other studies found the opposite results? Have non investigated this?

L274: Could it also have something to with bias – the one who did the registrations got better in investigating the udder for clinical signs? At the first visit the interview was also preformed so could the registrations have been done differently at the other visits (as then only the clinical examination was performed)?

L277-279: What have other studies found?

L280-282: Consider the wording in this sentence – it is not the farmers that are higher producers, but the cows on the farm…, please rewrite.

L285-288: Have other studies found differences in production system?

If you found that most of the herds had manage factors that in other studies have shown to increase the risk of mastitis I think it would be good to address that in the discussion – if all herds do “wrong” you will not find any statistical differences…

Conclusion

L291-292: Consider changing to …and subclinical mastitis and identified some risk factors on such farms.”

L292-293: Consider this conclusion – is it necessary to do more risk factor studies?

L292-295: I completely agree!!!

Round 2

Reviewer 1 Report

The manuscript “Risk Factors Associated With Mastitis in Smallholder Dairy Farms in Southeast Brazil” enters the topics of the journal, it is of great interest to readers. The paper is now much improved in its various parts and written with more care than the previous version, and in this way its general quality has increased and is in my opinion considerable for publication

The data presented by the authors are original and significant. The study is correctly designed and technically sounds. In general, the statistical analyses are performed with good technical standards. We authors conducted careful work which will attract the attention of a wide range of specialists focused on smallholder dairy farms.

I have reviewed the previous version of this paper and indicated several concerns that the authors should address. The authors revised the text according to my recommendations and, in general, fixed the concerns..

I suggest in line 68 : Highlight the usefulness of the study

“…Acting on these factors in a direct way it could be reduce the risk of clinical and subclinical mastitis in smallholders.”.

Line 73: maybe..."The 517 cows sampled come from 15 farms in the mesoregion of Piracicaba close to the University of São Paulo....."

I suggest clarifying throughout the article:: is the sample 517 cows, Quarter observations(n.)  or 10 farms?. I think that the farm is another factor, but maybe i'm wrong

Line 119-120 ... You say:

  Predictors with P-value <0.25 were then put forward for testing in multivariable models ... Why? Clarify please

Line 299:

I suggest:

This study in 517 cows, n quarter observations, and distributed in ten smallholder dairy farms in Brazil found a high prevalence of both clinical and subclinical mastitis.

Congratulations.

Reviewer 3 Report

Thank you for considering my suggestions to improve the quality of the manuscript. The manuscript has improved but I still have some considerations. First, I do recommend that you edit the manuscript, so it is grammatically corrected, now the wording is in the wrong order in too many sentences for me as a reviewer to accept, moreover the tempus is not correct everywhere. Then I have some specific comments:

Simple summary

L15: please changed detected to investigated

L15-17. Still not clear, please check the wording

L17-19: Please consider changing to “We identified some risk factors; increased parity, later stages of lactation, not milking clinical and subclinical mastitis cases last, lack of routine for cleaning the milk parlor, using dry-off treatment and optimized feeding before calving.” Parity is not a risk factor as younger cows have less CM/SCM and older have more, the risk factor is older cows/increased parity. Moreover, consider if optimized feed is the correct term for what you have observed.

Abstract

Shouldn’t the abstract include an aim?

L27-29: Why do you not present the OR and CI for these associations when you do it for the ones presented in the next sentence? Please be consistent, present them for all or none! You have not included the results for vaccine or season why not?

L33-34: This sentence is not necessary here, better to include your findings of vaccine and season.

Introduction

L45: please consider changing rate to cow level prevalence, a rate refers to eg. animal time is used in the nominator and neither Oliviera nor Busanello reports an incidence rate.

L57: please add an “and” after breed instead of a comma

L60-64: Good information!

Materials and Methods

L119: Please add a sentence with motivation for the categorization of DIM and season (visit date) and also explain what optimized feed is (was it an increased ratio of concentrates 15 days before calving?).

L123: How about interactions? Could there have been an interaction between parity and dim, parity and visit date?

L129-132: please add “such as” or something similar after Other management practices…, and check the grammar. Please add “after milking” after …keep cows standing… if that is what you refer to.

Table 1. Please include distribution of CM and SCM for each category, not the proportion of observations - that is not what you compare in the statistical analyses. I suggest that you instead of the two columns “Quarter observations” and “Proportion” you add four columns with CM yes/no and SCM yes/no – then it will be clear what proportions that have been used in the statistical analyses. Moreover, please align the numbers so that the singular, ten, hundreds and thousands are in the same position for the numbers below in the columns (using right margin) – this improves the readability of the tables.

Results

L154-158: Please add somewhere, here, or elsewhere, why these variables was not included in a model, as most of them usually are associated with the outcomes!

L158-159: Maybe this sentence should be removed to the discussion as it is not a result of your study? Also, please add information that you did not have information on BMSCC or IMI in these herds.

L162: Use decimal 1,165 and 4,567

L166: Please add the herd prevalence for CM and SCM, average and range. I also encourage you to add the information from your response 33, number of cows with both CM and SCM. It would also be interesting to know how many cows had CM/SCM in more than one quarter at the same occasion. Please also add the information from response 34, medium (range) of observations per cows.

L170-172: I am still a bit concerned by your categorization of DIM – could the significant difference in number of cases of CM/SCM in DIM 50-150 be due to an almost 4 times higher number of observations? How would the result be if you categorized using quartiles or deciles? If the reason is that you did not have exact DIM information, this should be stated in the M&M section in the statistic part where you should motivate your categorization.

L183: Please add information somewhere on how did the farmers register SCM so that they could know which to milk last? Was CMT testing commonly used?

L187: Please consider changing “relating” to “investigating”

L191: You do not mention visit date results, what was the risk season?

L194: You do not mention visit date results, what was the risk season?

L194-195: Do you mean that production system did not remain in any model but was offered to the model? Please clarify!

Table 2-4: I still believe that the interpretation for the general reader would increase if you added p-values in the tables, especially for those variables with more than two categories. Few veterinarians without research training will know how to interpret CI and be able to know if all categories were significantly different to the reference level, but most will understand a p-value. Cis also depends on n and data variability…

Table 3 and 4. OR is on the row of reference. I think the table outlay would improve if you removed the explanation after Category in the Category column and added reference in the odds ratio column, then the OR would come in the right row.

Table 4. Should it not say, “optimized feeding before calving”?

All tables – be consistent in number of decimal points used, now you sometimes have one, sometimes two and sometimes three… I suggest you stick to two!

Discussion

L268: please change “the poor…” to “a poorer”

L287: E. coli should be in italic

L288: You have to be more specific in what optimization of feed relates to

L294: Please also include that farm size, number of cows and milk production was not associated with CM and SCM, and a speculation on why.

Author Response

Please see the attachment."
